# *Siraitia grosvenorii* Extract Attenuates Airway Inflammation in a Murine Model of Chronic Obstructive Pulmonary Disease Induced by Cigarette Smoke and Lipopolysaccharide

**DOI:** 10.3390/nu15020468

**Published:** 2023-01-16

**Authors:** Mi-Sun Kim, Dong-Seon Kim, Heung Joo Yuk, Seung-Hyung Kim, Won-Kyung Yang, Geum Duck Park, Kyung Seok Kim, Woo Jung Ham, Yoon-Young Sung

**Affiliations:** 1Herbal Medicine Research Division, Korea Institute of Oriental Medicine, 1672 Yuseongdae-ro, Yuseong-gu, Daejeon 34054, Republic of Korea; 2Institute of Traditional Medicine and Bioscience, Daejeon University, 62 Daehak-ro, Dong-gu, Daejeon 34520, Republic of Korea; 3Suheung Research Center, Seongnam 13488, Republic of Korea

**Keywords:** SGE, chronic obstructive pulmonary disease (COPD), NF-κB, MAPK, neutrophils

## Abstract

We studied the activities of *Siraitia grosvenorii* extracts (SGE) on airway inflammation in a mouse model of chronic obstructive pulmonary disease (COPD) stimulated by cigarette smoke extract (CSE) and lipopolysaccharide (LPS), as well as in LPS-treated human bronchial epithelial cell line (BEAS-2B). SGE improved the viability of LPS-incubated BEAS-2B cells and inhibited the expression and production of inflammatory cytokines. SGE also attenuated the mitogen-activated protein kinase (MAPK)-nuclear factor-kappa B (NF-κB) signaling activated by LPS stimulation in BEAS-2B cells. In mice stimulated by CSE and LPS, we observed the infiltration of immune cells into the airway after COPD induction. SGE reduced the number of activated T cells, B cells, and neutrophils in bronchoalveolar fluid (BALF), lung tissue, mesenteric lymph node, and peripheral blood mononuclear cells, as well as inhibited infiltration into organs and mucus production. The secretion of cytokines in BALF and the expression level of pro-inflammatory cytokines, mucin 5AC, Transient receptor potential vanilloid 1, and Transient receptor potential ankyrin 1 in lung tissue were alleviated by SGE. In addition, to investigate the activity of SGE on expectoration, we evaluated phenol red secretions in the trachea of mice. SGE administration showed the effect of improving expectoration through an increase in phenol red secretion. Consequently, SGE attenuates the airway inflammatory response in CSE/LPS-stimulated COPD. These findings indicate that SGE may be a potential herbal candidate for the therapy of COPD.

## 1. Introduction

Chronic obstructive pulmonary disease (COPD), caused by inhaling irritants like cigarette smoke (CS) and air pollutants, destroys pulmonary tissue and parenchyma, leading to emphysema [1,2]. Chronic inhalation of irritants activates pattern recognition receptors, such as toll-like receptors, which activate airway epithelial cells with the release of mucus and an elevation in neutrophils and macrophages in the lung [3]. Neutrophils are momentous in the progress of mucus hypersecretion in COPD [4], which is characterized by progressive neutrophilic airway inflammation. The inflamed lung releases proteases such as matrix metalloprotease and elastase, which can lead to emphysema, and pro-inflammatory mediators like tumor necrosis factor (TNF)-α, interleukin (IL)-1β, IL-8, and IL-17 that attract inflammatory cells. In addition, activated neutrophils migrate to the pulmonary parenchyma [5,6]. Additionally, in COPD patients, increased CD8^+^ and CD4^+^ T lymphocytes infiltrate the adventitia layer of the pulmonary arteries and lung parenchyma [7]. 

Smoking is the main cause of COPD. Cigarette smoke (CS) exposure causes the infiltration of lymphocytes and neutrophils, which arouses the release of pro-inflammatory cytokines, such as TNF-α and IL-6, and airway tissue remodeling [8]. However, although the animal model using CS is an appropriate model for COPD, when CS is used alone, there is the limitation that induction of COPD is mild [9,10]. Lipopolysaccharide (LPS, a component of Gram-negative bacterial cell wall)-induced COPD model is a relatively easy model that can achieve COPD in a short time [11]. Combing the COPD model using CS with exposure to LPS increased COPD exacerbation and severity of emphysematous changes [12]. Therefore, the CS and LPS-induced COPD model can be a useful model of COPD. LPS generates pro-inflammatory cytokines and reactive oxygen species that increase the accumulation of inflammatory cells [13,14]. The interaction of LPS with Toll-like-Receptor (TLR) 4 induces the expression of inflammatory molecules such as IL-6, IL-8, and IL-1β, by bronchial epithelial cells, resulting in an immune reaction [15]. LPS incubation of human bronchial epithelial cells, which results in the release of inflammatory cytokines and MUC5AC, is controlled by the TLR4-NF-κB-MAPK pathway, indicating an inflammatory response [16,17].

Traditional herbal medicine exhibits a therapeutic effect on pulmonary disease by improving lung function. *Siraitia grosvenorii* (Luo Han Guo, Cucurbitaceae family) is a natural sweetener and medicinal herb used to treat stomach ailments, colds, sore throats, and various respiratory diseases including bronchitis and asthma [18], and *S. grosvenorii* extract (SGE) has antidiabetic, antitumor, antioxidative, and anti-inflammatory effects [19,20,21]. Mogroside V, one of the major components of SGE, reduces airway hyperresponsiveness and the level of inflammatory cytokines in bronchoalveolar lavage fluid (BALF) by inhibiting ovalbumin-induced NF-κB activation in asthmatic mice [22,23]. These data suggest that SGE may have curative activities on pulmonary disorders.

Recently, we reported the alleviation of asthma and atopic dermatitis by SGE [24,25]. However, the effects of SGE in COPD models and the signaling mechanisms underlying its effects have not been described. Here, we show that SGE inhibited airway inflammation in LPS-incubated BEAS-2B cells and CSE/LPS-induced COPD murine models. These findings demonstrate for the first time that SGE improves COPD. 

## 2. Materials and Methods

### 2.1. Preparation of Siraitia grosvenorii Extract

The alcohol extract of Luo Han Guo *Siraitia grosvenorii,* Mogron^®^, was provided by Suheung Co., Ltd. (Seongnam, Republic of Korea). Dried *S. grosvenorii* fruit was extracted with water. Following water extraction, *S. grosvenorii* residues were dried and extracted with 70% (*v*/*v*) ethanol. The extract was filtered, condensed, lyophilized, and kept at 4 °C. The yield of dried extract was about 5% (W/W). 

### 2.2. Cell Culture and Treatment

Human bronchial epithelial cell line (BEAS-2B) (ATCC, Manassas, VA, USA) was grown in Dulbecco’s modified Eagle’s medium (Gibco, New York, NY, USA) with 10% fetal bovine serum (FBS) and 1% penicillin–streptomycin at 37 °C in a 5% CO_2_ and 95% humidified atmosphere. BEAS-2B cells were stimulated with 20 μg/mL LPS (Sigma-Aldrich Co., St. Louis, MO, USA) to induce an immune response. 

### 2.3. Cell Viability Assay

The cells (5 × 10^3^ cells/well), plated in a 96-well plate, were stimulated with LPS at 0.1, 1, 5, 10, or 20 μg/mL, or SGE at 25, 50, 100, 200, or 400 μg/mL for 24 h. CCK-8 reagent (Cell Counting Kit-8 assay; Dojindo, Kumamoto, Japan) was treated to each well and stimulated for 2 h. Absorbance was evaluated at 450 nm.

### 2.4. RNA Extraction and Quantitative Analysis of mRNA Expression Levels

Total RNA prep HiGene™ kit (BIOFACT, Daejeon, Republic of Korea) was used to isolate total RNA from BEAS-2B cells, and total RNA (1 μg) was reverse transcribed to make cDNA using an iScript™ cDNA synthesis kit (Bio-rad, Hercules, CA, USA). Reverse transcriptase quantitative PCR (RT-qPCR) was carried out using the CFX Connect Real-Time PCR Detection System (Bio-Rad) with iTaq Universal SYBR Green Supermix. Transcripts of IL-1β, IL-6, IL-8, TNF-α, CXCL-1, IL-17, and IRAK-1 were expressed as ΔΔC_t_, normalized to the gene for actin. Primer sequences are in Appendix A.

Total RNA was extracted with the RNAsolB (Tel-Test, Friendswood, TX, USA) phenol–chloroform method to measure mRNA expression in COPD-induced lung tissue. Expression levels of TNF-α, MIP-2, CXCL-1, MUC5AC, Transient receptor potential cation channel subfamily V member 1 (TRPV1), and Transient receptor potential cation channel, subfamily A, member 1 (TRPA1) were determined using an Applied Biosystems 7500 Fast Real-Time PCR system (Foster, CA, USA) with a SYBR Green PCR Master mix and primers.

### 2.5. Immunoblot

The cells (6 × 10^5^ cells/well), seeded in a 6-well plate, were incubated or not with SGE for 24 h at 37 °C. Then, 20 μg/mL LPS was treated to the cells for 15 min or 1 h. BEAS-2B cell lysates were extracted by PRO-PREP buffer (Intron, Republic of Korea). Protein was electrophoresed by gel and then moved to membranes (Trans-blot Turbo transfer system, Bio-Rad). Blocking was executed using EzBlock Chemi buffer (ATTO, Koto, Japan) for 30 min and incubated with anti-NF-κB, anti-p-NF-κB, anti-IκB, anti-p-IκB, anti-JNK, anti-p-JNK, anti-p38, anti-p-p38, or anti-*β*-actin antibodies (Cell signaling Technology, Danvers, MA, USA) at 4 °C for 24 h. After washing, membranes were exposed to secondary antibodies (Cell Signaling) for 1 h. The band signal was recorded using chemiluminescence (GE Healthcare, Chicago, IL, USA).

### 2.6. Animal Model of COPD

Eight-week-old male Balb/c mice (Jackson Laboratory, Bar Harbor, ME, USA) were acclimatized for one week in an environment at 21 ± 2 °C, a humidity of 60 ± 10%, 150–300 lux illumination, and a 12 h light/dark cycle. This study was authorized by the Committee for Animal Welfare of Daejeon University (ethical approval code DJUARB2022-011), and all procedures were performed according to the Guide for the Care and Use of Laboratory Animals published by the National Institute of Health. After acclimatization, mice were separated into six groups (n = 12/group) at random: (1) normal, (2) COPD-control (CTL), (3) COPD-Dexa, (4) COPD-SGE-25 mg/kg, (5) COPD-SGE-50 mg/kg, or (6) COPD-SGE-100 mg/kg. COPD was caused by intranasal administration of mixture including 100 μg/mL LPS and 4 mg/mL cigarette smoke extract (CSE) to mice according to a previously described method [26]. Saline was administered to the normal group, and the mixture was administered to the remaining groups intranasally at 7-day intervals. Dexamethasone (3 mg/kg) and SGE (25, 50, 100 mg/kg) were given orally every day for 14 days. The dose of SGE used in these experiments was determined from a preliminary experiment based on our previous study on asthma [24].

### 2.7. Collection of Bronchoalveolar Lavage Fluid (BALF) and Lung Cells

Bronchoalveolar Lavage Fluid (BALF) was gained by cannulating the trachea of anesthetized mice and injecting cold DMEM into the lungs. The total number of cells was obtained by a hemocytometer; the cells were obtained by cytospin centrifugation and fixed, and cytological examination was performed through modified Diff-Quick staining. The number of differentiated cells was assessed using a cytospin slide. Lung tissue was minced using a sterile scalpel to isolate single cells and reacted in phosphate-buffered saline (PBS) with 2 mg/mL dispase and 1 mg/mL collagenase IV (Sigma-Aldrich), for 45 min at 37 °C. Then, the tissues were strained through a cell strainer (BD Falcon, Bedford, MA, USA) according to a previously described method [24]. 

### 2.8. Measurement of Cytokine

Human, IL-8, IL-1β, IL-6, TNF-α, IL-17, and CXC motif chemokine ligand 1 (CXCL-1) were measured from the cell culture supernatant using enzyme-linked immunosorbent assays (ELISA) kits (R&D Systems, Minneapolis, MN, USA) and the absorbance was also observed at 450 nm (SpectraMax; Molecular devices, San Jose, CA, USA). 

Collected BALF was prepared by cannulating the trachea of anesthetized mice and injecting 1 mL of 0.9% saline into the lungs. BALF was centrifuged at 1900 rpm for 15 min, and IL-17A, macrophage inflammatory protein-2 (MIP-2), CXCL1, and TNF-α cytokine levels were made in supernatants by ELISA kits (R&D).

### 2.9. Flow Cytometric Analysis (FACS)

Enzymatic digestion was performed for fluorescence-activated cell sorting (FACS) analysis of cells in lung tissue and mesenteric lymph nodes (MLN). The lung and MLN were removed from the mouse, washed three times, cut, and moved to a tube with Hank’s Balanced Salt Solution including 2% FBS and 1 mM Ethylene-diamine-tetraacetic acid (EDTA) for 30 min. The lung tissue was reacted with 1 mg/mL collagenase IV, filtered through a cell strainer, and centrifuged for 20 min. Collected cells from lung and MLN and BALF were rinsed twice and incubated with Immunoglobulin G (IgG) antibodies against CD8 (53-6.7, rat IgG2a), CD4 (RM4-5, rat IgG2a), CD21 (7G6, rat IgG2b), CD69 (H1.2F3, Hamster IgG1), B220 (RA3-6B2, rat IgG2a), SiglecF (1RNM44N, rat IgG2a), CD62L (MEL-14, rat IgG2a), Gr-1 (RB6-8C5, rat IgG2b), CD11b (M1/70, rat IgG2b), and CD44 (IM7, rat IgG2b). All fluorochrome-labeled monoclonal antibodies and IgG isotype controls were obtained from BD Biosciences (San Diego, CA, USA). Cells from the lung and BALF were reacted with fluorescein isothiocyanate (FITC)- and phycoerythrin (PE)-labeled monoclonal antibodies for 30 min. The stained cells were characterized using FACS Caliber (BD Biosciences) according to a previously described method [27]. 

### 2.10. Histological Analysis

Lung was infused with 1 mL of 10% formalin through the trachea, and the tissue was removed and immersed in a fixative for 24 h. The tissues were paraffinized, sectioned to a thickness of 6 μm, and stained with hematoxylin and eosin (H&E), Masson’s trichrome (M-T), and periodic acid-Schiff (PAS) (Sigma-Aldrich) stains. The stained tissue was observed by light microscopy for cell infiltration, fiber formation, and mucus production. The inflammation was measured on a subjective scale of 0–2 as previously described [28].

### 2.11. Immunohistofluorescent (IHF) Staining

Lung was mounted in optimal cutting temperature (OCT) embedding compound and kept at −20 °C. Using a cryostat microtome (Leica Microsystems, Wetzlar, Germany), tissue was sectioned to 20 μm thickness. The sectioned tissue was fixed in 4% formaldehyde with sucrose for 40 min, and the tissue was permeabilized in PBS with 0.5% nonyl phenoxypolyethoxylethanol (NP)-40. Tissue was blocked using 2.5% bovine serum albumin and horse serum for 16 h. Tissue was subjected to immunofluorescence staining by overnight incubation with antibodies against TNF-α, CXCL1, and Interleukin-1 receptor-associated kinase 1 (IRAK-1) at 4 °C. Tissues were washed, followed by treatment with a fluorescein-labeled secondary antibody for 2 h. Then, the nucleus was stained with Hoechst. The stained tissue was examined with a fluorescence microscope (Nikon Instruments Inc., Mississauga, ON, Canada) as previously described [24].

### 2.12. Phenol Red Secretion

This study was performed as described previously [29]. Seven-week-old male ICR mice (Orient-bio, Republic of Korea) were retained at a temperature of 22–24 °C and humidity of 50 ± 10% and acclimatized for 7 days in a controlled environment with adjustable cycle lighting. This study was authorized by the Committee for Animal Welfare of Daejeon University (ethical approval code DJUARB2022-027). For mucus secretion studies, the animals were randomly assigned to normal (vehicle-administrated), control (vehicle-administrated phenol red injection), Levosol (positive control; PharmGen Science Inc, Seoul, Republic of Korea), and 50 or 100 mg/kg of SGE (n = 10/group). SGE was administered orally once a day, at the same time for 3 days, and normal and control groups were administered saline (vehicle). After the last SGE administration, 5% phenol red was injected intraperitoneally 30 min later excluding normal group, and after 30 min, the trachea was transferred and immersed in 1 mL saline for 30 min to extract phenol red. Then, 0.1 mL Sodium hydroxide (NaOH) was treated, and absorbance was evaluated at 546 nm to quantify phenol red secretion.

### 2.13. Statistical Analysis

All the data are expressed as the mean± standard error of the mean (SEM). Statistical significance was carried out by one-way analysis of variance (ANOVA) by Duncan’s multiple comparison test using GraphPad Prism 7.0. 

## 3. Results

### 3.1. Effects of SGE on the Cytotoxicity of LPS-Stimulated BEAS-2B Cells

We determined the activities of SGE on LPS-induced inflammation in human bronchial epithelial cells incubated with a range of LPS doses. LPS at 10 or 20 μg/mL induced cytotoxicity in BEAS-2B cells compared to the untreated group (*p* < 0.01; Figure 1A, left). BEAS-2B cells incubated with various concentrations of SGE (0, 25, 50, 100, 200, or 400 μg/mL) showed a concentration-dependent increase in viability up to 400 μg/mL, which showed toxicity (Figure 1A, middle). For BEAS-2B cells stimulated with 20 μg/mL of LPS, SGE increased cell viability and alleviated cytotoxicity compared to the control (with dexamethasone as a positive control) (*p* < 0.05; Figure 1A, right).

### 3.2. Effects of SGE on the Secretion and Expression of Pro-Inflammatory Cytokines in LPS-Stimulated BEAS-2B Cells

The secretion of IL-6, IL-8, TNF-α, and MUC5AC, measured by ELISA, in LPS-stimulated BEAS-2B cells, decreased significantly with SGE incubation or with dexamethasone (*p* < 0.01, and *p* < 0.001; Figure 1B–E). We performed RT-qPCR to identify the activity of SGE on the expression of genes for pro-inflammatory cytokine and IRAK1 (Figure 1F). The expression of pro-inflammatory cytokines IL-1β, TNF-α, IL-8, IL-6, and IL-17 was significantly inhibited by SGE (*p* < 0.05). The expression of the genes for chemokine CXCL-1, related to neutrophil trafficking, and IRAK1, related to the expression of inflammatory genes, was also reduced by SGE ((*p* < 0.001).

### 3.3. Effects of SGE on the Inflammatory Response through the MAPK-NF-κB Signaling Pathway

To identify the mechanism of SGE inhibition of inflammation in LPS-treated bronchial epithelial cells, we measured MAPK and NF-κB signaling pathway proteins using immunoblots. The level of phosphorylated p65 and phosphorylated IκB increased in LPS-treated BEAS-2B cells (*p* < 0.05), whereas the ratio of phosphorylated p65 to p65 and phosphorylated IκB to IκB decreased after SGE treatment (*p* < 0.001; Figure 2A,B). In addition, phosphorylated JNK and phosphorylated p38, which are MAPK signaling-related proteins, decreased after SGE treatment (*p* < 0.01, and *p* < 0.001; Figure 2C,D).

### 3.4. Effects of SGE on Airway Inflammation in COPD-Induced Murine Model

As shown in Figure 3B–D, CSE/LPS exposures increased the total BALF cells, lung cells, and MLN cells (*p* < 0.001). In addition, the number of neutrophils in BALF was also increased by COPD induction (*p* < 0.001), which was shown by microscopic images of BALF cytospins (Figure 3E,F). On the other hand, administration of SGE (50 and 100 mg/kg) significantly decreased CSE/LPS-induced inflammatory cell recruitment (*p* < 0.05). Furthermore, SGE decreased the number of neutrophils in BALF (*p* < 0.01).

### 3.5. Effects of SGE on the Number of Airway Immune Cell Subtypes in BALF, Lung Tissues, and Mesenteric Lymph Node (MLN) of COPD-Induced Mice

Flow cytometric analysis on the immune cells in BALF in the COPD murine model demonstrated that the number of immune cells, such as activated T lymphocytes, and neutrophils increased in BALF by CSE/LPS stimulation compared to the normal group. In particular, the number of T lymphocytes (CD8^+^, CD4^+^, CD8^+^/CD69^+^) and neutrophils (Gr-1^+^/SiglecF^−^) was reduced significantly by SGE treatment (*p* < 0.05, *p* < 0.01, and *p* < 0.001; Table 1). Similarly, SGE treatment reduced the number of airway inflammatory cells from lung tissue that had been increased by COPD induction by CSE/LPS stimulation, including activated T cells (CD4^+^/CD69^+^, CD8^+^/CD69^+^, CD62L^−^/CD44^high+^), B cells (CD21^+^/B220^+^), and neutrophils (Gr-1^+^/SiglecF^−^) (Table 2, upper rows). The distribution of T cells in the MLN of COPD-induced mice revealed an increased number of CD4^+^, CD8^+^, CD4^+^/CD69^+^, CD62L^−^/CD44^high+^ cells compared to the normal group, but these cells decreased significantly with SGE treatment (*p* < 0.05, *p* < 0.01, and *p* < 0.001; Table 2, lower rows).

### 3.6. Effects of SGE on Cytokines in BALF and Expression Levels in Lung Tissue of COPD-Induced Mice

Cytokines TNF-α, IL-17, CXCL1, and MIP2, as measured by ELISA, increased in CSE/LPS-induced COPD mice but were decreased significantly by incubation with SGE (50 and 100 mg/kg) (*p* < 0.05, *p* < 0.01, and *p* < 0.001; Figure 4A–D). Expression of the genes encoding cytokines TNF-α, MIP-2, and CXCL1 in lung tissue, as measured by RT-qPCR, increased in COPD-induced mice (*p* < 0.05, *p* < 0.01, and *p* < 0.001; Figure 5A–C). The expression of the genes encoding TRPA1, MUC5AC, and TRPV1 was reduced significantly by SGE treatment (50 and 100 mg/kg) (*p* < 0.05, *p* < 0.01, and *p* < 0.001; Figure 5D–F). However, the expression of the gene for TRPV1 decreased significantly at 25 mg/kg SGE but not at higher concentrations (Figure 5E). 

### 3.7. Effects of SGE on the Histopathology of Lung Injury in COPD-Induced Mice

To determine the effects of SGE on lung damage in CSE/LPS-induced COPD, lung tissue stained with H&E, M-T, or PAS was observed by light microscopy. H&E staining revealed cell infiltration around small airway with thickening of the airway wall in the COPD lung tissue, whereas treatment with SGE reduced cell infiltration and decreased airway wall thickness (*p* < 0.01, and *p* < 0.001; Figure 6A,D). Increased fibrosis in COPD lung tissue was confirmed by M-T staining; however, treatment with SGE significantly reduced collagen fibers in the COPD lung tissue (Figure 6B). PAS staining also showed a reduction in SGE-treated mice of the mucus hypersecretion observed in the COPD lung tissue due to hyperplasia of goblet cells (Figure 6C). 

### 3.8. Effects of SGE on the Expression of TNF-α, CXCL-1, IRAK1 in COPD-Induced Mice

Immunofluorescence analysis showed that TNF-α, CXCL-1, and IRAK1 increased in the small airways in COPD-induced lung tissue. However, SGE treatment significantly reduced the expression of these inflammation factors (*p* < 0.05, *p* < 0.01, and *p* < 0.001; Figure 7).

### 3.9. Effects of SGE on Expectoration through Phenol Red Secretion

To investigate the activity of SGE on expectoration, we evaluated phenol red secretions in the trachea of ICR mice. Levosol relieves cough symptoms caused by bronchitis and sore throat [30]. We found that Levosol promoted mucus secretion, as demonstrated by the 5.3-fold rise in the secretion of phenol red compared to the control group. The groups treated with SGE at 50 and 100 mg/kg also demonstrated significant increases in the secretion of phenol red of 2.4- and 2.9-fold, respectively (*p* < 0.001; Figure 8). 

## 4. Discussion

*Siraitia grosvenorii* (swingle) is a traditional medicine used in China and Korea as an antitussive, for gastrointestinal disorders, and for sore throat [31]. In addition, SGE has antioxidant, antidiabetic, anticancer, and hepatoprotective effects [32,33,34,35]. We demonstrated previously the inhibitory effects of SGE on OVA-induced asthma and atopic dermatitis [24,25]; however, SGE has not been studied as a treatment for COPD. This disease is a form of emphysema featuring irreversible airflow resistance and is developed mainly by cigarette smoking or inhalation of fine dust or harmful gases. COPD, which causes serious health problems, is the third main cause of death in the world, and the number of patients continues to increase [36]. Here, we demonstrate that SGE has a beneficial effect on COPD, and we identify its mechanism of action.

It is well known that LPS binds to TLR4, thereby activating NF-κB and leading to the release of pro-inflammatory cytokines such as IL-6, IL-1β, and TNF-α, [37,38]. IL-17 promotes the secretion of GM-CSF and IL-6 from epithelial cells, thereby recruiting neutrophils to the airway, which results in the secretion of pro-inflammatory cytokines [39,40]. In BEAS-2B cells incubated with LPS or TNF-α, NF-κB and MAPK p38 are related to the production of CXCL-1 [41]. We demonstrated an inhibitory activity of SGE on LPS-stimulated lung inflammation in human bronchial epithelial cells, in which SGE inhibited LPS-induced cytotoxicity and the expression and release of inflammatory cytokines. In addition, SGE attenuated the MAPK- NF-κB cascade activated by LPS stimulation (Figure 2). Therefore, we demonstrated that the mechanism by which SGE alleviated lung inflammation was via the regulation of MAPK-NF-κB signaling.

Chronic obstructive lung disease is characterized by the remodeling of the airways associated with parenchymal destruction due to airway inflammation [42]. Immune cells such as CD8^+^, CD4^+^, and T helper cells are activated and increase in response to the deterioration associated with emphysema and respiratory restriction [43,44,45]. In particular, increased neutrophilic inflammation, a noticeable property of COPD, is associated with exacerbation of COPD [46,47]. Here, we found a rise in the number of inflammatory neutrophils and lymphocytes and an elevation in inflammatory cytokine levels (TNF-α, IL-17, CXCL-1, and MIP2) in the lung, BALF, and MLN in COPD-induced mice. This increase in inflammatory cells in the COPD-induced group was mitigated by SGE treatment, particularly the number of neutrophils in BALF. SGE also suppressed cytokines in BALF. An analysis of the subtypes of airway immune cells in MLN, BALF, and lung in COPD-induced mice through FACS confirmed that SGE inhibited the elevation in the number of activated T lymphocytes (CD4^+^, CD8^+^, CD4^+^/CD69^+^, CD8^+^/CD69^+^, CD62L^−^/CD44^high+^) and neutrophils (Gr-1^+^/siglecF^−^). Because neutrophils play a major mechanistic role in chronic inflammation of COPD, SGE likely alleviates COPD by preventing the increase in the number, activity, and infiltration of neutrophils. Taken together, our data suggest that SGE relieves pulmonary inflammation by reducing the number of infiltrated and activated immune cells in lung tissue, BALF, and MLN in the COPD mouse model.

The inflammatory cytokines CXCL1 MIP2, and TNF-α are secreted in response to continuous exposure to CS, mediating inflammation by neutrophils [48]. MUC5AC is a major mucin protein expressed in airway epithelial goblet cells [49], and TRPA1 and TRPV1 play a major part in the cough reflex [50]. Analysis of the transcripts for the genes encoding TNF-α, MIP2, CXCL-1, MUC5AC, TRPV1, and TRPA1 in the lung of COPD mice revealed a significant increase in the CSE/LPS-stimulated control groups that was inhibited by SGE. Thus, we suggest that SGE regulates the inflammation of COPD via its inhibitory action.

Characteristics of COPD include abnormal epithelial remodeling, basal cell hyperplasia, and fibrosis around small airways, along with goblet cell metaplasia [51,52]. In our COPD mouse model, histopathology showed the infiltration of immune cells and airway remodeling in lung tissue. We investigated that SGE ameliorated the accumulation of inflammatory cells around the small airway, fibrosis, mucus secretion, and the increase in goblet cell size that were associated with COPD. Thus, SGE protected from COPD-induced lung tissue damage by inhibiting these pathological changes. We also confirmed by immunofluorescence analysis of lung tissue in the COPD-induced mouse model that TNF-α, CXCL1, and IRAK1 were dose-dependently reduced by SGE. IRAK1 plays a major function in mediating the release of TNF-α, MIP-2, CXCL1, and IL-17 cytokines along with the MAPK pathway in airway inflammation, which leads to alveolar wall thickening and accumulation of collagen fibers [53]. Therefore, SGE may prevent airway inflammation by suppressing the secretion of inflammatory cytokines via regulating IRAK1.

In COPD, expectorant remedy is commonly used to alleviate airway stenosis, prevent recurrent infection and exacerbation, promote mucus activity, and improve airflow and lung function [54]. We found that the secretion of phenol red from the trachea was increased in mice treated with Levosol or SGE, supporting an expectoration effect of SGE. 

Our study demonstrated that CSE/LPS-induced COPD and airway inflammation was inhibited by SGE treatment. The most effective dose of SGE in mice is 100 mg/kg and based on body surface area, the practical human dose of SGE is about 500 mg/day (adult). However, the human dosage can differ depending on the treatment frequency, age, and gender. Females are more susceptible to the lung-damaging activities of cigarette smoking than males due to sex hormone differences; therefore, the therapeutic approach of COPD can differ by gender [55,56]. A limitation of our study is that it did not demonstrate an effect according to sex differences. Nevertheless, our results suggest that SGE could be a promising agent to treat respiratory disease, such as COPD. These findings demonstrate for the first time that SGE improves COPD. From these results, a human clinical study of SGE will proceed in the future. 

## 5. Conclusions

We demonstrated the protective effect of *S. grosvenorii* extract in CSE/LPS-induced COPD murine models. COPD increased immune cell infiltration around small airways and inflammatory cytokine secretion, but SGE alleviated neutrophilic inflammation and lung tissue damage by modulating MAPK-NF-kB signaling. Therefore, SGE is an herbal agent with the potential to treat COPD. 

## Figures and Tables

**Figure 1 nutrients-15-00468-f001:**
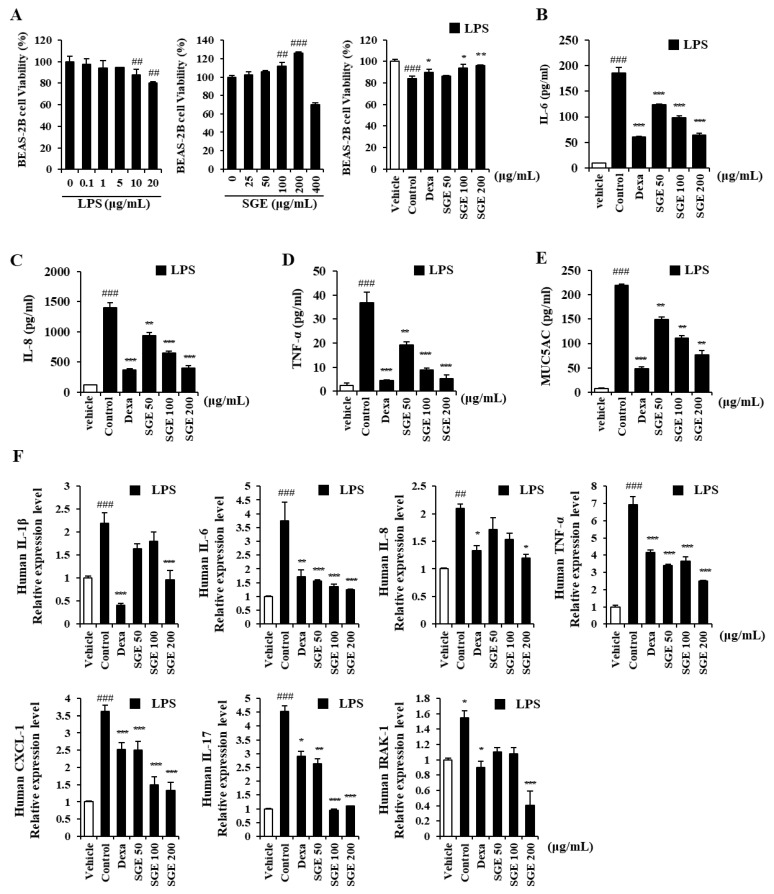
Effects of *S. grosvenorii* extract (SGE) on cytotoxicity and pro-inflammatory cytokine production in LPS-induced BEAS-2B cells. (**A**) Cytotoxicity of LPS and SGE in BEAS-2B cells was measured using the CCK-8 assay. (**B**–**E**) BEAS-2B cells were treated with LPS at 20 μg/mL and SGE at 50, 100, or 200 mg/mL for 24 h. Cytokines (IL-6, IL-8, TNF-α) and MUC5AC were measured using ELISA. (**F**) The relative expression of mRNA of the genes for IL-1β, IL-6, IL-8, TNF-α, IL-17, CXCL1, and IRAK1 in the LPS-stimulated cells. The data are presented as means ± SEM for three independent experiments. ## *p* < 0.01, ### *p* < 0.001 compared to the vehicle group. * *p* < 0.05, ** *p* < 0.01, *** *p* < 0.001 compared to the control group. Dexa, dexamethasone.

**Figure 2 nutrients-15-00468-f002:**
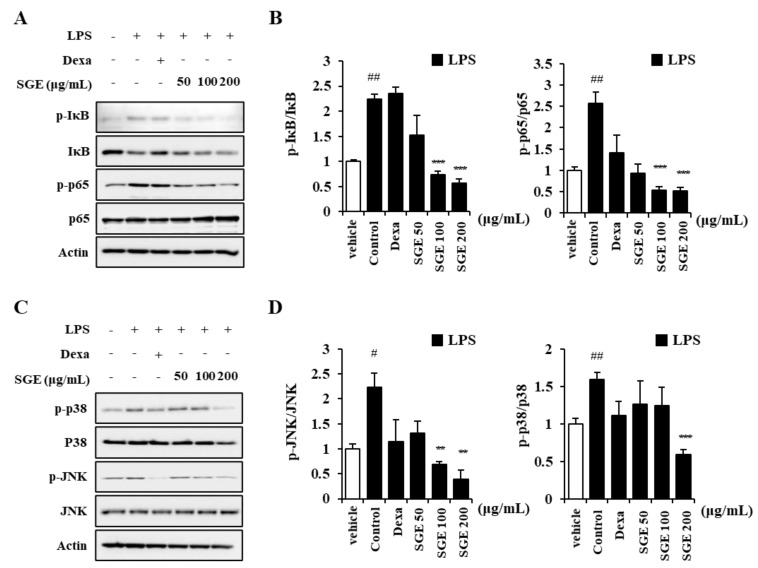
Effects of SGE on LPS-induced MAPK-NF-κB pathway in BEAS-2B cells. (**A**,**B**) The ratio of p-IκB to IκB and p-p65 to p65 was determined by pretreatment of BEAS-2B cells with SGE (50, 100, or 200 μg/mL) for 24 h, followed by stimulation with 20 μg/mL of LPS for 1 h and then immunoblotting. (**C**,**D**) The ratio of p-JNK to JNK and p-p38 to p38 was determined by pretreatment of BEAS-2B cells with SGE (50, 100, or 200 μg/mL) for 24 h, followed by stimulation with LPS for 1 h and then immunoblotting. Raw data of Western blot are described in the Appendix A. The data are expressed as means ± SEM of three independent experiments. # *p* < 0.05, ## *p* < 0.01 compared to the vehicle group. ** *p* < 0.01, *** *p* < 0.001 compared to the control group. Dexa, dexamethasone.

**Figure 3 nutrients-15-00468-f003:**
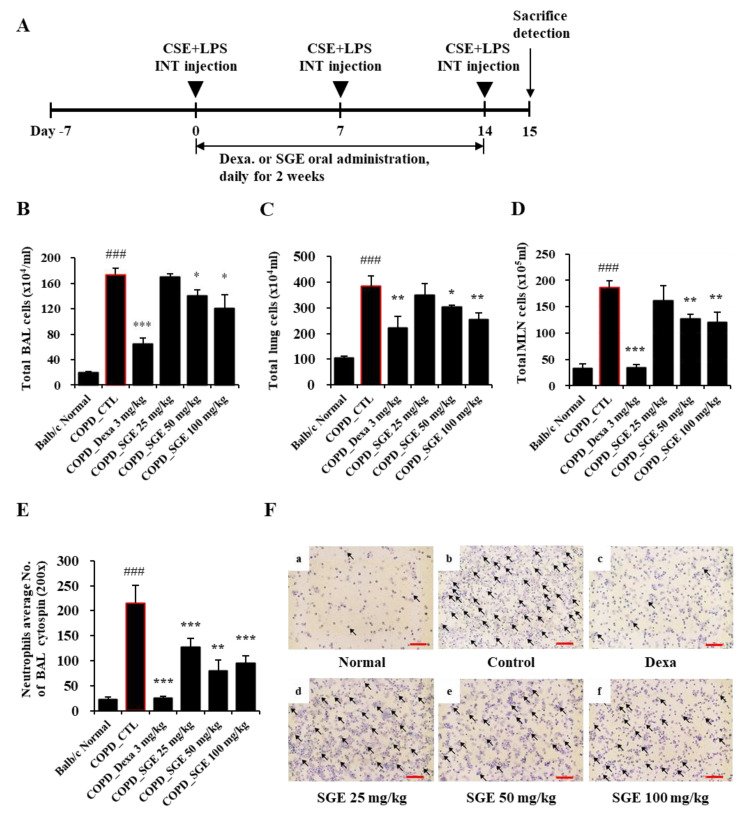
Effects of SGE on the number of airway immune cells and neutrophilic airway inflammation in CSE/LPS-induced COPD murine model. (**A**) Schematic diagram of CSE/LPS-induced COPD model in mice. (**B**) The total number of cells in COPD-induced BALF in each treatment group. (**C**) The total number of cells in the lungs of COPD-induced mice in each treatment group. (**D**) The total number of cells in COPD-induced MLN in each treatment group. (**E**) The number of neutrophils in BALF was counted using a hemocytometer. (**F**) Photomicrograph of BALF cytospins from COPD model mice treated with CSE/LPS (magnification, 200×) by differential leukocyte staining. Neutrophils are indicated by arrows. Scale bar = 100 µm. The data are presented as means ± SEM (n = 6 mice) for individual mice. ### *p* < 0.001 compared to normal, and * *p* < 0.05, ** *p* < 0.01, and *** *p* < 0.001 compared to control. Dexa, dexamethasone.

**Figure 4 nutrients-15-00468-f004:**
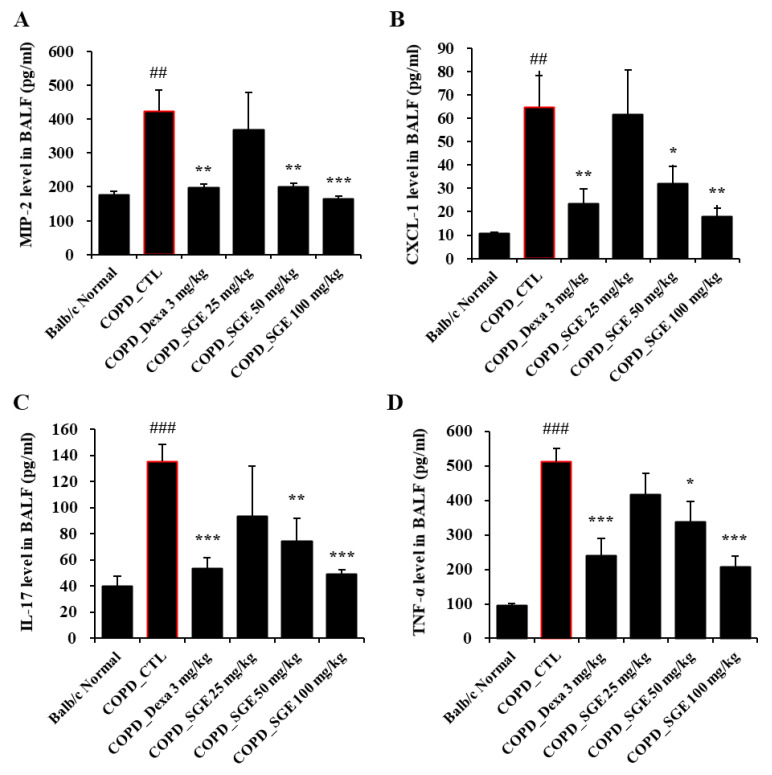
Effects of SGE on cytokine production in BALF of COPD-induced mice. Production of (**A**) MIP2, (**B**) CXCL1, (**C**) IL-17, (**D**), and TNF-α in BALF of COPD-induced mice was measured by ELISA. The data are presented as means ± SEM (n = 6 mice) for individual mice. ## *p* < 0.01, ### *p* < 0.001 compared to normal, and * *p* < 0.05, ** *p* < 0.01, and *** *p* < 0.001 compared to control. Dexa, dexamethasone.

**Figure 5 nutrients-15-00468-f005:**
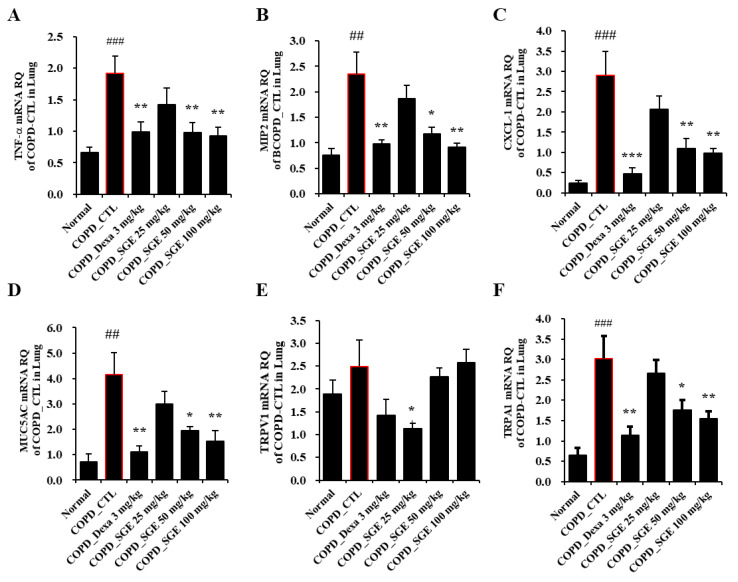
Effects of SGE on the expression of genes for TNF-α, MIP2, CXCL1, MUC5AC, TRPV1, and TRPA1 in lung tissue of COPD-induced mice. Expression of (**A**) TNF-α, (**B**) MIP2, (**C**) CXCL-1, (**D**) MUC5AC, (**E**) TRPV1, and (**F**) TRPA1 genes was determined by RT-qPCR in lung tissue of COPD-induced mice. The data are presented as means ± SEM (n = 6 mice) for individual mice. ## *p* < 0.01, ### *p* < 0.001 compared to normal, and * *p* < 0.05, ** *p* < 0.01, and *** *p* < 0.001 compared to control. Dexa, dexamethasone.

**Figure 6 nutrients-15-00468-f006:**
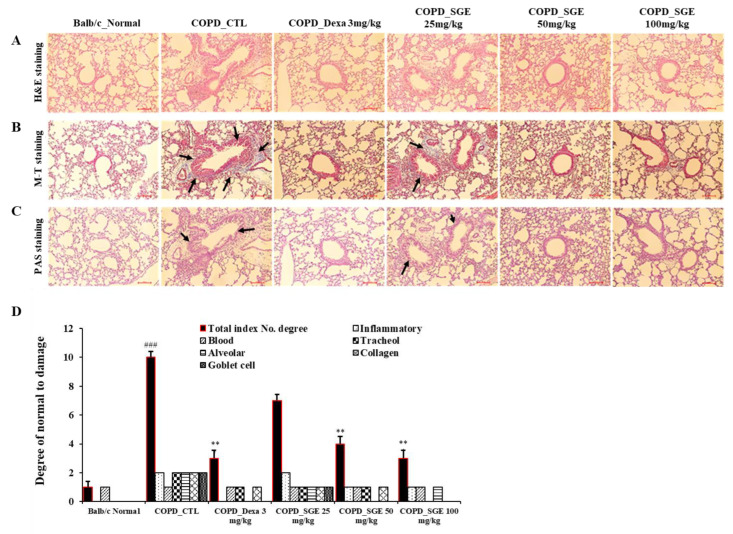
Effects of SGE on histopathological markers for lung tissue of COPD-induced mice. Lung tissue was sectioned in paraffin and stained for histological analysis. (**A**) H&E staining showing immune cell infiltration and the morphology of small airways. (**B**) M-T staining showing collagen fibers increased by pulmonary fibrosis. Black arrows indicate blue-dyed collagen fibers. (**C**) PAS staining showing hyperplasia of goblet cells in small airways. Black arrows indicate goblet cells. Scale bar = 100 µm. (**D**) Total index number were quantitated. The data are presented as means ± SEM (n = 6 mice) for individual mice. ### *p* < 0.001 compared to normal, and ** *p* < 0.01 compared to control. Dexa, dexamethasone.

**Figure 7 nutrients-15-00468-f007:**
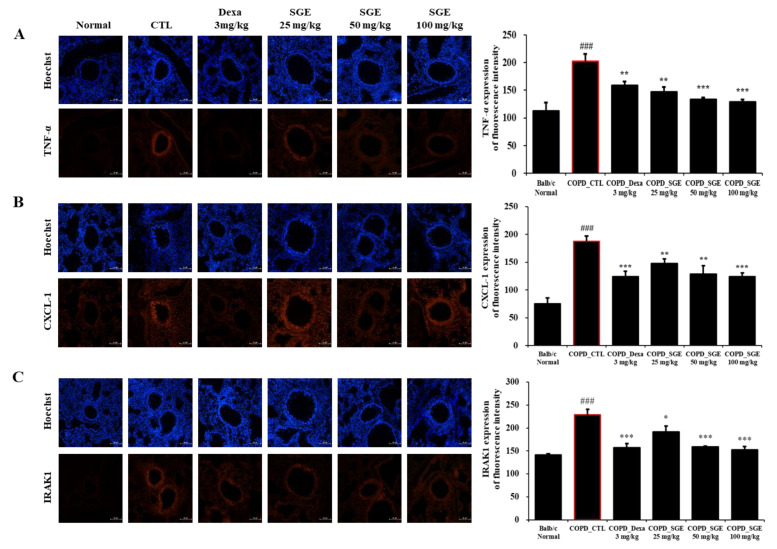
Effect of SGE on TNF-α, CXCL1, IRAK1 using immunofluorescence analysis of lung tissue from COPD-induced mice. Sectioned COPD-induced mouse lung tissue was stained with (**A**) TNF-α, (**B**) CXCL-1, or (**C**) IRAK1 antibodies and visualized as red fluorescence. Hoechst was used as a nuclear stain with blue fluorescence. Fluorescence for each factor is shown in a quantitative bar graph. Scale bar = 100 µm. The data are presented as means ± SEM (n = 6 mice) for individual mice. ### *p* < 0.001 compared to normal, and * *p* < 0.05, ** *p* < 0.01, and *** *p* < 0.001 compared to control. Dexa, dexamethasone.

**Figure 8 nutrients-15-00468-f008:**
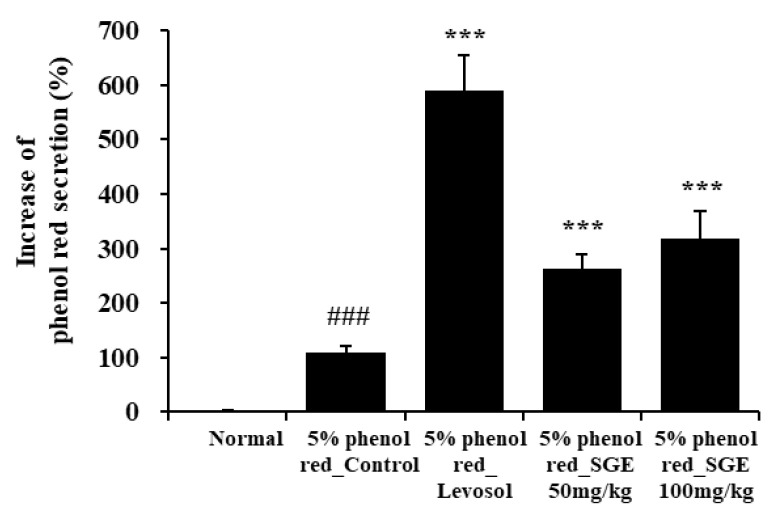
Effect of SGE on phenol red secretion in ICR mice. The amount of phenol red secretion in the airways was measured by injecting 5% phenol red into mice treated with Levosol or SGE for 3 days. The data are presented as means ± SEM (n = 10 mice) for individual mice. ### *p* < 0.001 compared to the normal group, and *** *p* < 0.001 compared to the control group.

**Table 1 nutrients-15-00468-t001:** The quantification of airway immune cell subtypes by FACS analysis in BALF from SGE-treated mice with COPD-induced inflammation.

Cell Phenotypes in BALF		COPD-Induced Inflammation Murine Model (Absolute No.)					
		Balb/c Normal	COPD_CTL	COPD_Dexa 3 mg/kg	COPD_ SGE 25 mg/kg	COPD_ SGE 50 mg/kg	COPD_ SGE 100 mg/kg
Lymphocyte (×10^4^ cells)	BAL	2.23 ± 0.54	8.99 ± 2.58 ^##^	4.07 ± 1.37	12.31 ± 2.18	11.86 ± 2.23	21.97 ± 14.46
Neutrophils (×10^4^ cells)		8.94 ± 1.16	156.49 ± 13.30 ^###^	58.11 ± 12.02 ***	151.16 ± 5.72	124.37 ± 11.39	110.11 ± 29.31 *
Eosinophils (×10^4^ cells)		0.46 ± 0.15	3.42 ± 0.33 ^###^	1.30 ± 0.65 **	4.82 ± 1.36	3.05 ± 0.32	2.52 ± 0.89
CD4^+^ (×10^4^ cells)		0.29 ± 0.15	41.29 ± 5.05 ^###^	9.21 ± 1.95 ***	28.48 ± 2.21 ^*^	17.10 ± 3.60 **	16.62 ± 5.98 **
CD8^+^ (×10^4^ cells)		0.09 ± 0.04	57.14 ± 7.85 ^###^	10.59 ± 5.39 ***	26.40 ± 1.55 ^**^	17.33 ± 1.83 ***	20.66 ± 6.47 **
CD4^+^CD69^+^ (×10^4^ cells)		2.23 ± 0.54	8.99 ± 2.58 ^##^	4.07 ± 1.37	12.31 ± 2.18	9.84 ± 0.38	6.71 ± 0.83
CD8^+^CD69^+^ (×10^4^ cells)		0.04 ± 0.02	9.06 ± 0.99 ^###^	1.08 ± 0.59 ***	5.05 ± 0.83 **	3.04 ± 0.39 ***	2.68 ± 0.46 ***
Gr-1^+^SiglecF^−^ (×10^4^ cells)		0.72 ± 0.25	141.73 ± 15.48 ^###^	23.73 ± 21.89 ***	124.61 ± 5.48	58.64 ± 28.49 **	62.37 ± 11.94 ***

The data are presented as means ± SEM (n = 8). ## *p* < 0.01, ### *p* < 0.001 versus the normal group, and * *p* < 0.05, ** *p* < 0.01, *** *p* < 0.001 versus the CTL (control) group, as determined by analyses of variance (ANOVA) followed by Duncan’s multiple range tests.

**Table 2 nutrients-15-00468-t002:** The quantification of airway immune cell subtypes by FACS analysis in lung and MLN of SGE-treated COPD-induced inflammation model.

Cell Phenotypes in Lung		COPD-Induced Inflammation Murine Model (Absolute No.)					
		Balb/c Normal	COPD_CTL	COPD_Dexa 3 mg/kg	COPD_ SGE 25 mg/kg	COPD_ SGE 50 mg/kg	COPD_ SGE 100 mg/kg
Lymphocyte (×10^4^ cells)	Lung	40.21 ± 16.30	37.76 ± 12.20	31.07 ± 11.57	33.75 ± 6.67	38.03 ± 7.60	42.86 ± 6.19
Neutrophils (×10^4^ cells)		30.22 ± 11.00	264.38 ± 72.27 ^##^	96.16 ± 13.07 *	195.59 ± 45.95	176.57 ± 45.40	147.33 ± 28.13 *
Eosinophils (×10^4^ cells)		2.80 ± 0.87	18.40 ± 3.10 ^###^	6.45 ± 0.26 **	12.84 ± 2.37	12.27 ± 4.67	12.46 ± 6.54
CD4^+^ (×10^4^ cells)		32.52 ± 12.11	88.80 ± 24.11 ^#^	39.80 ± 8.10	77.73 ± 21.51	82.84 ± 22.23	81.19 ± 14.85
CD8^+^ (×10^4^ cells)		13.17 ± 5.00	50.46 ± 14.64 ^#^	25.06 ± 5.20	28.81 ± 6.26	32.50 ± 9.46	30.37 ± 4.19
CD4^+^CD69^+^ (×10^4^ cells)		0.38 ± 0.13	21.50 ± 4.78 ^###^	2.85 ± 0.39 **	11.60 ± 2.96	9.05 ± 2.46 ^*^	7.30 ± 2.29 **
CD8^+^CD69^+^ (×10^4^ cells)		0.70 ± 0.21	6.41 ± 1.77 ^##^	1.63 ± 0.34 **	3.65 ± 0.52	3.70 ± 1.52	2.82 ± 0.75 *
CD62L^−^CD44^high+^ (×10^4^ cells)		6.61 ± 2.15	130.71 ± 30.63 ^###^	26.05 ± 2.06 **	86.05 ± 18.87	77.27 ± 21.16	67.04 ± 14.47
CD21^+^B220^+^ (×10^4^ cells)		0.43 ± 0.21	5.36 ± 1.73 ^##^	0.95 ± 0.14 **	1.70 ± 0.37 *	1.26 ± 0.52 *	1.34 ± 0.35 *
Gr-1^+^SiglecF^−^ (×10^4^ cells)		6.41 ± 2.84	196.26 ± 57.67 ^##^	45.90 ± 5.76 **	139.72 ± 37.24	116.79 ± 37.24	57.13 ± 18.69 *
**Cell phenotypes** **in MLN**		**COPD-induced inflammation murine model (Absolute No.)**					
		**Balb/c Normal**	**COPD_CTL**	**COPD_Dexa** **3 mg/kg**	**COPD_** **SGE 25 mg/kg**	**COPD_** **SGE 50 mg/kg**	**COPD_** **SGE 100 mg/kg**
CD4^+^ (×10^4^ cells)	MLN	14.77 ± 5.38	70.76 ± 8.96 ^###^	9.82 ± 4.15	58.00 ± 12.36	35.56 ± 13.31 *	49.98 ± 10.71
CD8^+^ (×10^4^ cells)		6.08 ± 1.35	37.53 ± 5.27 ^###^	7.06 ± 3.89 ***	26.11 ± 0.34 *	18.72 ± 7.86 *	25.75 ± 4.08
CD4^+^CD69^+^ (×10^4^ cells)		2.93 ± 1.22	8.38 ± 1.46 ^##^	0.90 ± 0.16 ***	6.16 ± 2.41	5.71 ± 1.56	2.51 ± 0.68 **
CD62L^−^CD44^high+^ (×10^4^ cells)		4.55 ± 2.44	17.18 ± 2.51 ^##^	6.54 ± 1.15 **	11.12 ± 1.33 *	5.77 ± 1.67 **	8.81 ± 3.08 *

The data are presented as means ± SEM (n = 8). # *p* < 0.05, ## *p* < 0.01, ### *p* < 0.001 versus the normal group, and * *p* < 0.05, ** *p* < 0.01, *** *p* < 0.001 versus the CTL (control) group, as determined by analyses of variance (ANOVA) followed by Duncan’s multiple range tests.

## Data Availability

The data presented in this study are available in this article.

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
