# Peer review of "Siraitia grosvenorii* Extract Attenuates Airway Inflammation in a Murine Model of Chronic Obstructive Pulmonary Disease Induced by Cigarette Smoke and Lipopolysaccharide"

_nutrients, 2023, doi:10.3390/nu15020468_

Round 1

Reviewer 1 Report

1. Introduction: in paragraph 2, CS/LPS-induced in vivo model of COPD should be mentioned.

2. Yield of SGE after extraction and lyophilization should be mentioned.

3. A scheme of experimental schedule should be added to clarify the time point for induction of disease model and drug treatment.

4. Why did the authors select 25, 50, 100 mg/kg of SGE for treatment in the animal study? Toxicological data or related references should be added.

5. Why were different mouse strains (Balb/c and ICR) used for different experiments, not same strain?

6. Figure 2: NF-kB is a transcription factor, the expression of NF-kB should be measured in nuclear fraction rather than total cell lysates.

Why only the expression of p38 and JNK was evaluated, not ERK MAPK?

7. Scale bar length should be indicated in figure legend of staining results.

Reviewer 2 Report

Overall, the manuscript needs to be rewritten as the current structure and flow is not optimal. The results are confusing to follow and the Discussion lacks discussion. Methods were not well cited. Below are some major concerns to be addressed:

Abstract:

1)      Line 26: how do you measure ‘expectoration’ in mice?

2)      A lot of abbreviations used which requires definition, cannot assumed the readers know what they mean.

3)      Overall, the abstract flow and structure needs to be rewritten to improve readability.

Introduction:

1)      Unclear why LPS and cigarette mouse models were chosen?

2)      Overall, the flow and structure of the Introduction needs to be rewritten. The paragraphs don’t link well together. The points within the paragraphs also don’t flow well.

3)      More information about SGE is required. Why SGE was investigated?

4)      Justification and novelty of the study needs to be stronger.

Methods:

1)      Section 2.1-2.5, 2.7-2.12: there were no references cited in these methods? Are these methods all novel and never done before? For example, how was the concentrations for LPS and SGE were chosen for the in vitro experiments?

2)      Were the in vitro experiments performed at least three times?

3)      Which genes were chosen for RT-PCR? What were their primers sequence?

4)      A lot of abbreviations were not defined in the first use.

5)      Never start a sentence with an abbreviation.

6)      Why were male mice chosen and not female mice?

7)      How was the concentration for the drugs chosen in the mouse experiments?

8)      More information on the cigarette and LPS mouse protocol is required or at least a figure of the mouse experiment timeline to improve readers understanding.

9)      Section 2.12: was there animal ethics approval for this experiment? Why was a different strain of mice used? And why only male mice?

10)   What’s the difference between Normal group vs Control group? Please be more specific.

11)   Sample size for the different sections needs to be clearer.

Results:

1)      Figure 1A should be 3 separate figures as their statistics were not done together.

2)      Exact P-values should be written in-text.

3)      ‘Normal’ is not the right term to use.

4)      Figure 1 and 2: unclear what the ‘normal’ group is? And why was only ‘Control’ compared with ‘normal’?

5)      Figure 1F should be separate figures as their statistics were not done together.

6)      The ‘W’ in Western blot is capital letter.

7)      Unclear why some targets were measured based on their gene expression and some with their protein expression?

8)      Results section should only have results stated, do not include background information.

9)      Figure 3E: labelling is required, unsure what the cells are.

10)   Figure 6: more labelling is required, and were these results measure qualitatively? Were there any definite quantitative measurements? Cannot eyeball histology without scientific measurements.

Discussion:

1)      Never start a sentence with an abbreviation.

2)      The first two paragraphs should merge together as they are repetitive.

3)      What are the limitations of the study and what are the novelty about this study?

4)      Any comment about sex difference? What would be the practical dose and frequency of SGE is required for human? More discussion on clinical impact is required.

5)      Overall, the Discussion needs to be rewritten as the Discussion should be about discussing your results with previous results, and not just a reiteration of your results.
